# FlexTraj: Image-to-Video Generation with Flexible Point Trajectory Control

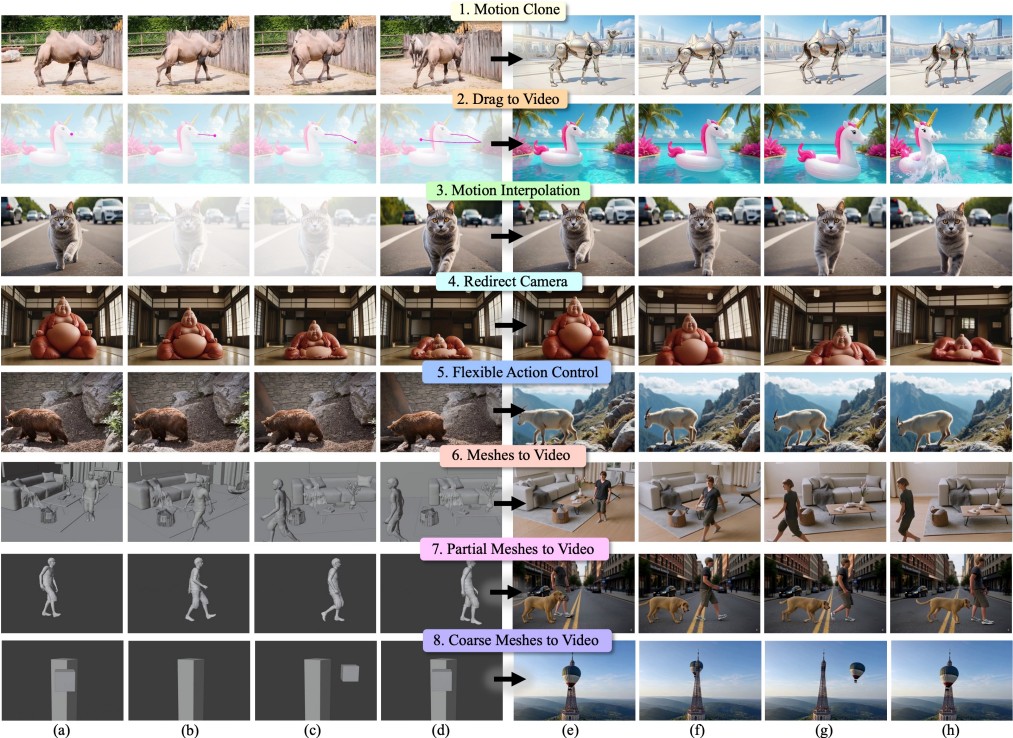

Figure 1: Examples synthesized by FlexTraj. FlexTraj supports multi-granularity trajectory control, including **dense** (e.g., *motion clone*, *camera redirection*, *mesh-to-video*), **spatially sparse** (e.g., *drag-to-video*, *partial mesh-to-video*), and **temporally sparse** (e.g., *motion interpolation—only provide motion on temporally sparse frames*) settings. Also it allows **unaligned** control (e.g., *flexible action control*, *coarse mesh-to-video*). Panels a–d are source and e–h are generated frames.

## ABSTRACT

We present FlexTraj, a framework for image-to-video generation with flexible point trajectory control. FlexTraj introduces a unified point-based motion representation that encodes each point with a segmentation ID, a temporally consistent trajectory ID, and an optional color channel for appearance cues, enabling both dense and sparse trajectory control. Instead of injecting trajectory conditions into the video generator through token concatenation or ControlNet, FlexTraj employs an efficient sequence-concatenation scheme that achieves faster convergence, stronger controllability, and more efficient inference, while maintaining robustness under unaligned conditions. To train such a unified point trajectory-controlled video generator, FlexTraj adopts an annealing training strategy that gradually reduces reliance on complete supervision and aligned condition. Experimental results demonstrate that FlexTraj enables multi-granularity, alignment-agnostic trajectory control for video generation, supporting various applications such as motion cloning, drag-based image-to-video, motion interpolation, camera redirection, flexible action control and mesh animations.

# 1 INTRODUCTION

While recent diffusion-based models (e.g., Sora (Brooks et al., 2024), CogVideoX (Yang et al., 2024b), Wan (Wan et al., 2025)) have achieved impressive visual quality in video generation, controllability remains an open challenge. To enable controllability, prior methods have proposed different types of conditioning signals such as depth maps (Lin et al., 2024; Jiang et al., 2025), edges (Xing et al., 2024; Guo et al., 2024), bounding boxes (Wang et al., 2024a; Ma et al., 2024), or masks (Li et al., 2025; Xing et al., 2025), which provide task-specific guidance but remain limited to a single control granularity. In contrast, point-trajectory control can naturally express a continuous spectrum of granularity through adjustable sampling density. Yet this capability has been underexplored: most prior methods (Yin et al., 2023; Wu et al., 2024) are limited to 2D dragging trajectories, while those extending into 3D remain are restricted to either sparse (Wang et al., 2025a) or dense control (Gu et al., 2025). A recent work (Geng et al., 2025) attempts to unify sparse and dense control by densifying sparse signals at inference with manually crafted templates. However, this design is fundamentally limited in both precision and flexibility, as the templates are hand-engineered and the model is not explicitly trained to accommodate diverse input conditions. Moreover, these approaches typically assume strict structural alignment between the input condition and the first source frame, which greatly constrains their practical applicability.

Motivated by these limitations, we propose FlexTraj, a unified framework for multi-granularity and alignment-agnostic point-trajectory control. We represent motion as a sequence of annotated 3D points, each associated with three attributes: a segmentation ID to distinguish object instances, a trajectory ID to ensure temporal correspondence across frames, and an optional[1] color attribute to encode appearance cues. These annotated points are projected into pixel space to form two conditioning videos: (i) an ID-coded video, combining segmentation and trajectory IDs, and (ii) a Color-cue video, encoding per-point colors. Both are directly processed by the pretrained video VAE to produce compact condition tokens. By varying the point sampling density, it enables controllability across different granularities, while projecting all densities into conditioning videos ensures a unified encoding. Furthermore, spatial shifts simulate unaligned inputs, allowing the model to adapt to misaligned conditions.

Injecting the above condition tokens into the generative model is not straightforward. A simple approach is to employ a ControlNet-style injector (Zhang et al., 2023); however, it shows suboptimal controllability on DiT backbones (Zhang et al., 2025a; Tan et al., 2024) and implicitly enforces structural alignment, making it unsuitable for unaligned inputs (e.g., last row in Fig.1). To address this, we propose an efficient sequence-concatenation strategy, which concatenates condition tokens with text embeddings and noisy latent tokens while using LoRA adaptation. Through attention interactions rather than direct addition, our method better accommodates unaligned inputs. Following EasyControl (Zhang et al., 2025a), we further introduce a causal mask that restricts condition tokens to attend only to themselves within the attention layers, ensuring self-consistency without interfering with other modalities. This design also enables KV caching at inference for faster computation.

As illustrated in Fig. 1, our framework can accommodate diverse conditions, including dense (e.g., 1st row), spatially sparse (e.g., 2nd row), temporally sparse (e.g., 3rd row), and unaligned inputs (e.g., 8th row). Training a single model to capture all these scenarios is non-trivial, and simply mixing tasks by random sampling during training leads to suboptimal results, as the model struggles to balance dense, sparse, and unaligned supervision simultaneously. To address this, we adopt a density and alignment annealing training curriculum: the model is first trained on complete conditions then gradually exposed to incomplete conditions, and finally to unaligned cases. This progression enables the model to generalize smoothly across levels of sparsity and alignment.

Thanks to our flexibility, our FlexTraj can naturally support different types of user groups. For general users, FlexTraj directly supports creative tasks such as video motion transfer, camera redirecting, motion interpolation, and drag-based image to video. For professional CG users, FlexTraj significantly reduces production effort. It can transform untextured renders into photorealistic videos in the dense setting, propagate motion from a partially rigged mesh to animate the full scene in the sparse setting, and synthesize plausible videos from coarse meshes in the unaligned setting (e.g., use simple primitives like cubes to provide motion guidance.) Experiments demonstrate that FlexTraj

---

[1]The color attribute is optional: it is specified for cases like camera redirecting or when explicitly mentioned; otherwise, it is omitted, and the model can handle both settings.

can not only generates high-quality, temporally coherent videos, but also provides flexible control across diverse scenarios. We summarize main contributions as follows:

- We introduce FlexTraj, the first framework to support multi-granularity and alignment-agnostic trajectory control, enabling varieties of applications shown in Fig.1.
- We propose a point trajectory representation that encodes segmentation IDs, temporal IDs, and optional color attributes in a unified manner, thereby establishing a general and flexible paradigm for controllable video generation.
- We present an efficient sequence-concatenation strategy that not only accelerates convergence and enhances controllability compared to ControlNet-style architectures, but also inherently supports unaligned conditions.
- We develop an annealing training curriculum that transitions from complete to incomplete and unaligned conditions, improving generalization across diverse user inputs.

## 2 RELATED WORK

**Video Diffusion**. Video diffusion models (VDMs) have advanced rapidly, extending the success of image diffusion into the temporal domain. Early efforts such as AnimateDiff (Guo et al., 2023) added temporal layers to pre-trained image diffusion models, while VideoCrafter (Chen et al., 2024) and SVD (Blattmann et al., 2023) improved fidelity and consistency. A major milestone came with Sora (Brooks et al., 2024), which demonstrated that combining a DiT (Peebles & Xie, 2023) backbone with massive training corpora can generate videos lasting up to a minute while maintaining high fidelity. Building on this trend, open-source models including CogVideoX (Yang et al., 2024b), WAN (Wan et al., 2025) and Hunyuan (Weijie Kong & Jie Jiang, 2024) follow the DiT paradigm and employ spatiotemporal VAEs to jointly compress space and time, but they still rely primarily on text or image prompts, leaving fine-grained motion controllability an open challenge.

**Non-Point Based Control.** Early controllable video generation approaches (Chen et al., 2023; Lin et al., 2024) primarily extended ControlNet (Zhang et al., 2023) conditions into the temporal domain. These conditions included structural cues such as depth maps, sketches, and edges (Lin et al., 2024; Jiang et al., 2025; Xing et al., 2024; Guo et al., 2024), which guided frame-by-frame synthesis. Human poses (Chang et al., 2023; Hu, 2024) also became popular, enabling dance or action videos with skeleton sequences. Beyond dense conditions, other approaches support sparse inputs such as bounding boxes or camera poses. For example, methods like Direct-A-Video (Yang et al., 2024a) or MotionCanva (Xing et al., 2025) allow users to direct camera and object movements. Extending into 3D, Cinemaster (Wang et al., 2025b) and 3DTrajMaster (Xiao et al., 2024a) leverage 3D bounding boxes or 6D pose sequences for motion control. A recent attempt, MagicMotion (Li et al., 2025), uses masks and boxes for dense and sparse control, but the two remain discrete rather than continuous; it only supports object-level 2D motion without part-level or 3D control.

**Point-Trajectory Based Control.** A range of 2D-based methods have been proposed to translate user-specified strokes or trajectories into video motion. DragNuwa (Yin et al., 2023) and MotionCtrl (Wang et al., 2023) map sparse strokes to Gaussian-maps, while DragAnything (Wu et al., 2024) further combines entity representations for entity-level control. Motion-I2V (Shi et al., 2024) and MoFA (Niu et al., 2024) introduce a two-stage pipeline that first predicts motion from strokes and then generates videos conditioned on predicted motion, but this requires two separate models and adds complexity. More recently, ToRA (Zhang et al., 2025b) leverages a DiT backbone (Peebles & Xie, 2023) to achieve state-of-the-art results, while Go-with-the-Flow (Burgert et al., 2025) explores dense trajectory control through optical flow warping. Motion-Prompt (Geng et al., 2025) further supports both sparse and dense control by densifying sparse signals with templates, but these are hand-crafted and the model is not trained for diverse conditions. Despite these advances, existing methods remain limited in handling 3D phenomena such as occlusions and rotations.

Some recent works have started to explore 3D-aware control. LeviTor (Wang et al., 2025a) clusters segmentation masks into sparse points and augments them with depth, but the points lack correspondence and the U-Net–based architecture constrains performance. DAS (Gu et al., 2025) enables 3D-aware dense control by propagating colors initialized in the first frame to enforce temporal consistency, but since point identities are fixed at initialization, it cannot represent newly emerging objects. Moreover, all prior approaches assume trajectories are aligned with the first frame, which

Figure 2: Overview of the FlexTraj framework. Given 3D-tracking points annotated with TrackID, SegID, and optional Color, users can sparsify or shift trajectories to define spatially sparse, temporally sparse, or unaligned controls. These modified trajectories are projected into condition videos (ID-coded and color-cue) and combined with the first frame and text prompt as inputs to a video diffusion model via efficient sequence-concatenation.

restricts their applicability. To address these limitations, we propose FlexTraj, a multi-granularity, alignment-agnostic trajectory control framework for image-to-video generation.

## 3 METHOD

We show the framework of FlexTraj in Fig. 2. We begin by encoding either a real-world video or a CG scene into a 3D trajectory representation and obtain a ID-coded condition video and a Color-cue condition video §3.1. We then employ a pre-trained VAE to encode both video conditions into token representations, which are subsequently fused and injected into the video generator via a efficient sequence concatenation strategy to conditionally generate the final videos §3.2. To effectively training such a framework, we adopt a density and alignment annealing training strategy that gradually reduces reliance on complete conditioning, thereby enabling robust controllability even under sparse or unaligned motion supervision §3.3.

### 3.1 TRAJECTORY REPRESENTATION

A good point trajectory representation should be flexible, expressive, and capable of preserving correspondence. Yet existing approaches either lack correspondence (Wu et al., 2024; Yin et al., 2023) or are not comprehensive (Gu et al., 2025; Geng et al., 2025), as described in §A.3. These shortcomings motivate FlexTraj to define a set of point trajectories in the following form:

$$\mathcal{P} = \left\{ p_i^t = (x_i^t, y_i^t, z_i^t, s_i, u_i, a_i) \mid i = 1, \ldots, N, \ t = 1, \ldots, T \right\}, \tag{1}$$

where $(x_i^t, y_i^t, z_i^t)$ denotes the 3D location of point $i$ at frame $t$. Each point $p_i^t$ carries three attributes: a segmentation identifier $s_i \in \mathbb{N}$ distinguishing object instances, a trajectory identifier $u_i \in \mathbb{N}$ ensuring temporal correspondence, and an optional vector $a_i \in \mathbb{R}^3$ providing color cues.

From the annotated trajectories, we render two condition videos: the ID-coded video $V_{\text{ID}}$, which encodes segmentation identifiers in the red channel and trajectory identifiers in the green–blue channels, and the color-cue video $V_{\text{Color}}$, which records the optional color vector $a_i$ projected onto the image plane. Together, these videos provide a compact encoding of object identity, temporal consistency, and visual appearance, which will serve as conditioning inputs in the next section.

### 3.2 CONDITION INJECTING CONTROL

After obtaining the trajectory conditions, the next step is to incorporate these conditions into the generative model. An intuitive approach is to use ControlNet (Zhang et al., 2023), as illustrated in Fig.3-(a). However, this method is suboptimal on the DiT backbones (Zhang et al., 2025a) and encourages strict structural alignment. To accommodate unaligned conditions, an alternative approach is direct sequence concatenation, as shown in Fig.3-(b), but this leads to significant computational complexity during training. To address this, we propose an efficient sequence-concatenation, which

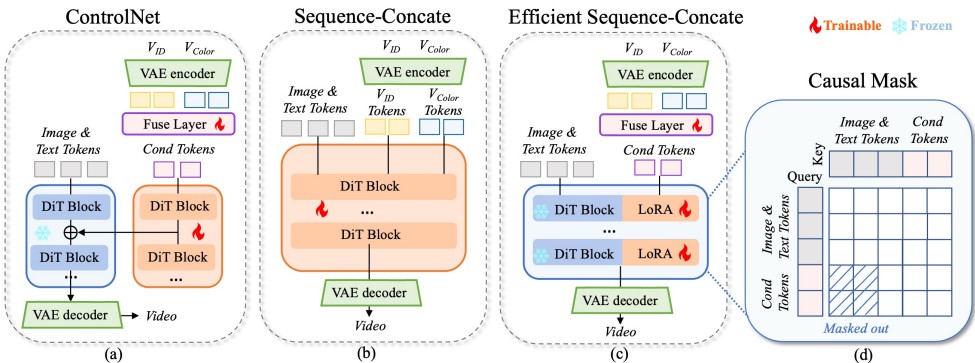

Figure 3: Comparison of condition-injection frameworks. (a) ControlNet-Style condition injection. (b) Sequence-Concatenation condition injection. (c) Our Efficient Sequence-Concatenation with LoRA and masked attention. (d) Causal mask.

achieves both flexibility and efficiency, as shown in Fig.3-(c). In the following, we outline three key components of this design: condition token fusion, LoRA adaptation with its training objective, and causal masking with KV caching for efficient inference.

**Condition Token Fusion.** We first encode the ID-coded video $V_{\text{ID}}$ and the color-cue video $V_{\text{Color}}$ into latent representations using a pretrained VAE encoder from CogVideoX (Yang et al., 2024b):

$$Z_{\text{ID}} = \text{VAE}(V_{\text{ID}}), \quad Z_{\text{Color}} = \text{VAE}(V_{\text{Color}}). \tag{2}$$

We then fuse both condition tokens as follows:

$$Z_c = Z_{\text{ID}} + W Z_{\text{Color}}, \tag{3}$$

where $W$ is a zero-initialized linear projection ensuring that appearance cues are integrated without overwriting structural information. Then, $Z_c$ is concatenated with the noise $Z_n$ and text tokens $Z_t$, forming the unified input sequence $Z$, defined as

$$Z = [Z_n \,;\, Z_t \,;\, Z_c]. \tag{4}$$

It should be noted that the condition tokens $Z_c$ are assigned the same positional encoding as the noise tokens $Z_n$, thereby preserving spatial alignment cues with the image tokens.

**LoRA-Adaptation and Training Objective.** We adopt Low-Rank Adaptation (LoRA (Hu et al., 2022)) to efficiently finetune the model while freezing the base diffusion transformer. To preserve the pretrained generative ability, LoRA is applied in query–key–value projections and only enabled when processing condition tokens. Formally, if $(Q_c, K_c, V_c)$ denote the original projections for condition tokens, the adapted forms are

$$Q_c' = Q_c + \Delta Q_c, \quad K_c' = K_c + \Delta K_c, \quad V_c' = V_c + \Delta V_c, \tag{5}$$

where $\Delta Q_c, \Delta K_c, \Delta V_c$ are low-rank updates learned during training. The model is trained using the standard diffusion objective (Ho et al., 2020). Given a clean video latent $x_0$, a noise sample $\epsilon \sim \mathcal{N}(0, I)$, and a timestep $t$, the noisy latent is defined as: $x_t = \sqrt{\alpha_t}\, x_0 + \sqrt{1 - \alpha_t}\, \epsilon$, where $\alpha_t$ is the variance schedule. The denoising network $\epsilon_\theta(x_t, t, Z)$, conditioned on the concatenated tokens $Z$, is optimized by

$$\mathcal{L}_{\text{diff}} = \mathbb{E}_{x_0, \epsilon, t} \left[ \| \epsilon - \epsilon_\theta(x_t, t, Z) \|_2^2 \right]. \tag{6}$$

**Causal Mask and KV Cache.** Following EasyControl (Zhang et al., 2025a), we apply a causal attention mask that prevents condition tokens from attending to noise or text tokens, while still allowing the latter to query information from conditions. The mask is defined as:

$$M_{ij} = \begin{cases} -\infty, & i \in Z_c,\, j \in (Z_n \cup Z_t), \\ 0, & \text{otherwise,} \end{cases} \tag{7}$$

Since condition tokens remain fixed across timesteps, their key–value projections $(K_c, V_c)$ can be computed once at $t = 0$ and cached for reuse, which substantially reduces inference cost without altering the conditioning effect.

### 3.3 DENSITY AND ALIGNMENT ANNEALING TRAINING STRATEGY

Training such a unified framework that supports multi-granularity control is not trivial, especially when incorporating unaligned conditions. Initially, we attempted to mix tasks by randomly sampling conditions of different types, but the results were unsatisfactory. We attribute this to the expanded parameter search space: densely aligned inputs offer strong determinism, while unaligned inputs require greater flexibility. This disparity creates a challenge for stable convergence, as the model struggles to balance the two contrasting demands.

To address this issue, we introduce an annealing training curriculum consisting of four stages. 1) In the first stage, the model is trained under the most deterministic conditions: a dense and aligned setting where both ID-coded and color-cue videos are consistently provided, ensuring rich information and rapid convergence. 2) In the second stage, supervision remains dense, but the color-cue video is randomly omitted with probability $p_c$. Despite partially reduced input signals, the deterministic nature of the dense setting ensure stable convergence. 3) After the model stabilizes with dense inputs, we gradually introduce spatial and temporal sparsity. Spatial sparsity is simulated either by randomly discarding trajectories or by segment-wise dropping, retaining only $p_s$ of the trajectories. Temporal sparsity is introduced in parallel by retaining only $p_t$ of the frames, selected either uniformly across the sequence or randomly. 4) In the final stage, the model is trained under unaligned conditions, where point trajectories are shifted relative to the input frame and a reduced learning rate is applied to mitigate catastrophic forgetting of capabilities acquired in earlier stages. To increase variation, we also synthesize unaligned trajectory pairs from CG scenes.

## 4 EXPERIMENTS

### 4.1 EXPERIMENT SETTINGS

**Dataset.** For training, we construct a dataset comprising approximately 40K real-world videos from VideoPainter (Bian et al., 2025) and 2.5K dance videos from HumanVid (Wang et al., 2024b). In addition, we synthesize around 5K videos using 3D meshes and animations collected from Mixamo (Blackman, 2014), incorporating identical poses across different characters to construct unaligned pairs. For evaluation, we adopt DAVIS (Pont-Tuset et al., 2017) and configure it for four evaluated tasks: dense, spatially sparse, temporally sparse, and unaligned. We also curate FlexBench, which contains 40 videos to showcase our method's strengths. See §A.1 for details.

**Baseline.** Since no existing baseline uniformly supports all evaluated tasks (dense, spatially sparse, temporally sparse, and unaligned), we select the most suitable methods for each task. In total, we compare our method with six baselines: four point-trajectory–based approaches (DAS (Gu et al., 2025), ToRA (Zhang et al., 2025b), LeviTor (Wang et al., 2025a), and Go-with-the-Flow (Burgert et al., 2025)), one box/mask-trajectory–based approach (MagicMotion (Li et al., 2025)), and one temporally sparse edge-map–based approach (SparseCtrl (Guo et al., 2024)). For each baseline, we format the inputs as required and generate results using their released code and pretrained models. The specific tasks supported by each method are summarized in Tab. 2.

**Metrics.** We evaluate our results using standard metrics: Fréchet Video Distance (FVD (Unterthiner et al., 2018)) and Frame Consistency (Esser et al., 2023) for video quality, and Trajectory Error (TrajError (Wu et al., 2024)) and Trajectory Similarity (TrajSIM (Pondaven et al., 2025)) for motion controllability. Details are provided in §A.1.

### 4.2 QUALITATIVE COMPARISON

**Dense Control.** We first evaluate the dense control on applications such as motion clone and mesh-to-video, comparing against MagicMotion (mask-trajectory-based) (Li et al., 2025), Go-with-the-Flow (Burgert et al., 2025), and DAS (Gu et al., 2025). As shown in Fig. 4, the 2D-based methods MagicMotion and Go-with-the-Flow struggle with fine details, as evident in the girl's head orientation and the jumping person's limbs. DAS also fails to capture the girl's head movement that appears in later frames, since its trajectory representation cannot accommodate newly emerging points. In contrast, our method, benefiting from a robust point representation and 3D awareness, achieves the best alignment with the source frames.

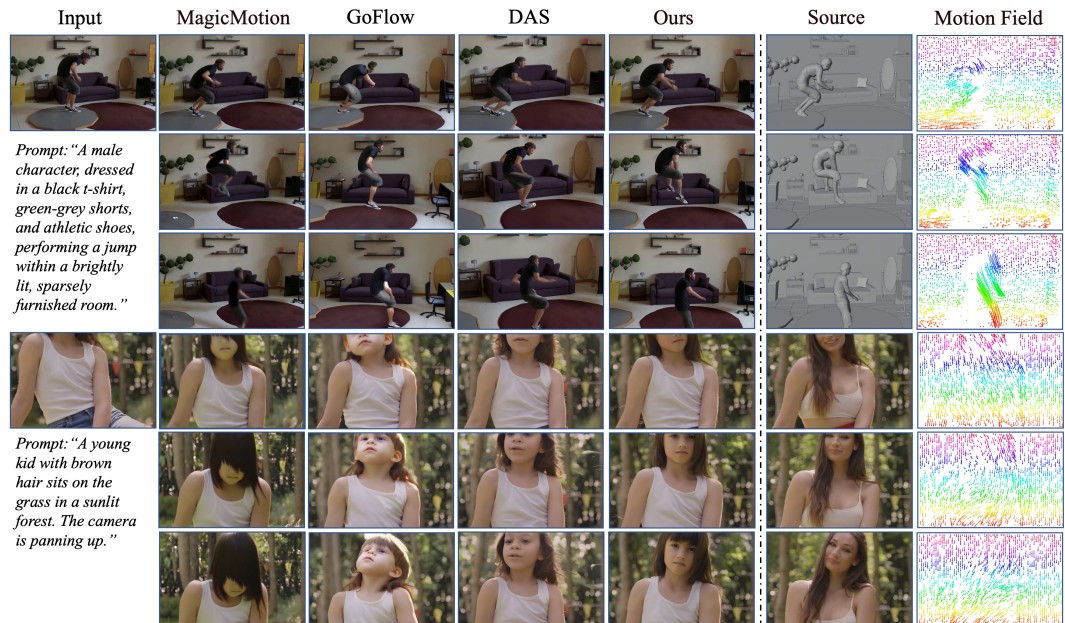

Figure 4: Qualitative comparison on dense control. MagicMotion (Li et al., 2025) and Go-with-the-Flow (Burgert et al., 2025) struggle with fine-grained details; DAS (Gu et al., 2025) fails to handle newly emerging points, whereas our method closely follows the source motion.

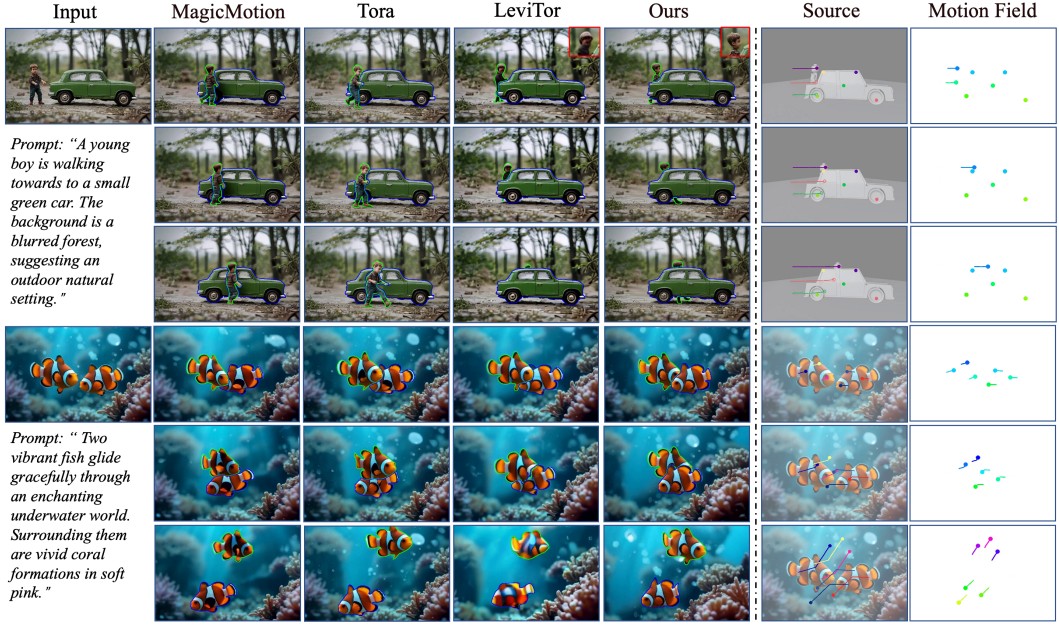

Figure 5: Qualitative comparison on spatially sparse control. The subject outlined in green is occluded by the subject outlined in blue. 2D-based methods (MagicMotion (Li et al., 2025), ToRA (Zhang et al., 2025b)) fail in handling occlusion, U-Net-based method LeviTor (Wang et al., 2025a) introduces artifacts, while ours accurately captures occlusion with high visual fidelity.

**Spatially Sparse Control.** Next, we evaluate spatially sparse control, comparing our method with two point-trajectory approaches (ToRA (Zhang et al., 2025b) and LeviTor (Wang et al., 2025a)) and the box-trajectory method MagicMotion (Li et al., 2025). As shown in Fig. 5, MagicMotion and ToRA both fail to handle occlusion correctly, as they are 2D-based and lack 3D awareness. LeviTor, while 3D-aware, produces noticeable artifacts (such as the distorted boy's face and unnatural render-

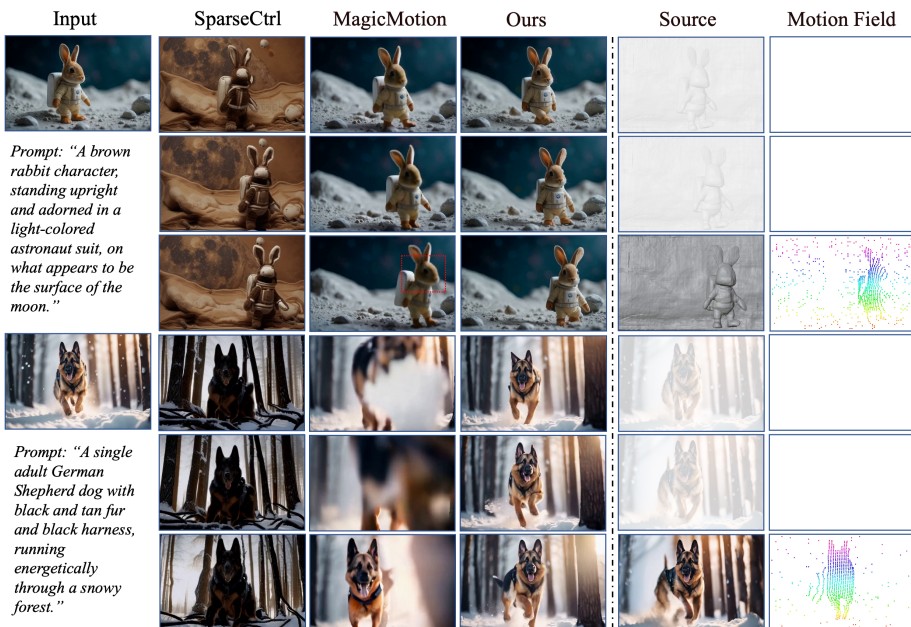

Figure 6: Qualitative comparison on temporally sparse control. SparseCtrl (Guo et al., 2024) yields unsatisfactory results, while MagicMotion (Li et al., 2025) shows weak alignment and blurriness. Our method aligns with the anchor-frame motion and generates coherent in-between frames.

ing of the fish), reflecting limitations of its U-Net–based architecture. In contrast, our method, with its robust 3D-aware representation, faithfully captures occlusion while preserving visual fidelity.

**Temporally Sparse Control.** Then, we evaluate temporally sparse control against two baselines: the sketch-based SparseCtrl (Guo et al., 2024) and the box-trajectory-based MagicMotion (Li et al., 2025), where the motion is specified only on a few anchor frames and the remaining frames are generated freely. As shown in Fig. 6, SparseCtrl, limited by its U-Net architecture, produces unsatisfactory results. MagicMotion exhibits weak alignment, with the dog disappearing and reappearing incorrectly and the rabbit's head appearing blurry. In contrast, our method generates coherent intermediate frames that remain well aligned with the motion indicated by the anchor frames.

**Unaligned Control.** Finally, we evaluate our method under unaligned conditions, comparing it with two dense point–trajectory approaches: DAS (Gu et al., 2025) and Go-with-the-Flow (Burgert et al., 2025). As shown in Fig. 7, DAS produces red artifacts around the subject, reflecting the strict alignment bias of ControlNet (Zhang et al., 2023). Go-with-the-Flow yields implausible results, such as a squirrel with a distorted tail and a bag suddenly appearing on a boy's back, due to mismatches between the input image and the motion field. In contrast, our method shows greater flexibility by referencing motion cues from the input without relying on strict spatial alignment.

## 4.3 QUANTITATIVE COMPARISON

We present a comprehensive quantitative evaluation in Tab. 1. Our approach consistently demonstrates superior trajectory control, achieving the lowest trajectory error (TrajErr (Wu et al., 2024)) and highest trajectory similarity (TrajSIM (Pondaven et al., 2025)) across four evaluated tasks. In addition to precise trajectory alignment, our method delivers competitive or improved video quality in terms of FVD (Unterthiner et al., 2018) and Frame Consistency (Esser et al., 2023), highlighting its ability to balance controllability with generation fidelity.

## 4.4 ABLATION

We evaluate the effectiveness of our core design choices (trajectory representation, condition injection, and the annealing training strategy) with qualitative examples and conclude with quantitative results. Details are provided in §A.4.

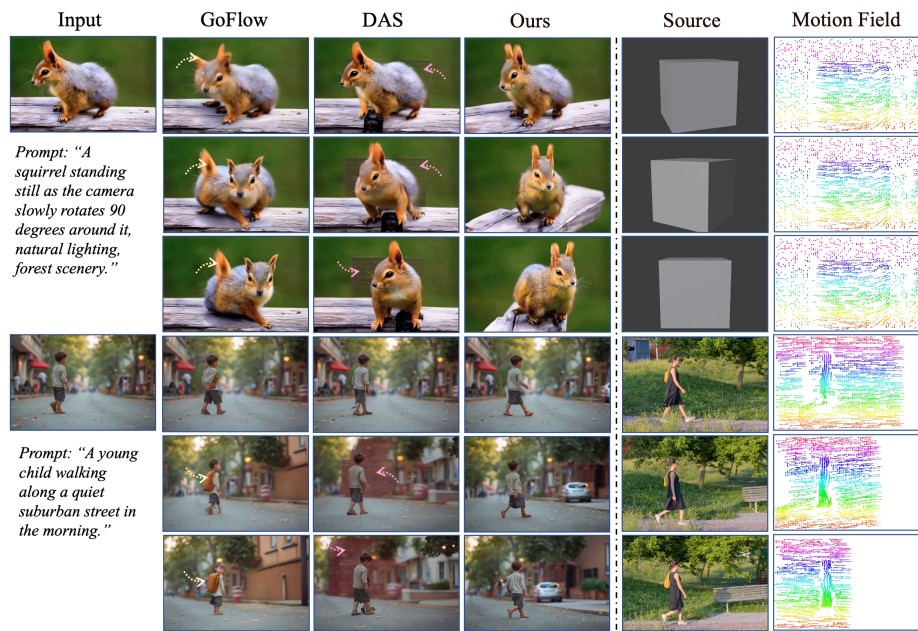

| Input | GoFlow | DAS | Ours | Source | Motion Field |
|---|---|---|---|---|---|

Figure 7: Qualitative comparison on unaligned control. DAS (Gu et al., 2025) introduces artifacts (red artifacts around the subject) from strict alignment, while Go-with-the-Flow (Burgert et al., 2025) produces implausible results. Our method flexibly follows input motion.

Table 1: Quantitative comparison with baseline methods. Our approach consistently outperforms all baselines in trajectory control, achieving the lowest TrajErr and highest TrajSIM. In terms of overall video quality (FVD and Consistency), our method attains competitive or superior performance.

| Task | Method | DAVIS | | | FlexBench | | |
|---|---|---|---|---|---|---|---|
| | | FVD↓ | Consistency↑ | TrajErr↓ | FVD↓ | Consistency↑ | TrajErr↓ |
| Dense | DAS | 714.3 | **0.981** | 0.029 | **1338.8** | 0.982 | 0.039 |
| | GoFlow | 793.1 | 0.975 | 0.044 | 1560.7 | **0.977** | 0.026 |
| | MagicM | 705.3 | 0.980 | 0.116 | 1621.0 | 0.985 | 0.062 |
| | Ours | **532.4** | 0.979 | **0.017** | 1397.8 | 0.982 | **0.014** |
| Spatially Sparse | ToRA | 1233.3 | 0.974 | 0.058 | 1210.2 | 0.988 | 0.037 |
| | Levitor | 1337.3 | 0.951 | 0.050 | 1944.2 | 0.970 | 0.044 |
| | MagicM | 860.5 | 0.980 | 0.080 | 978.1 | 0.988 | 0.045 |
| | Ours | **710.4** | **0.980** | **0.025** | **851.6** | **0.991** | **0.017** |
| Temporally Sparse | SparCtrl | 2533.4 | 0.967 | 0.087 | 2949.8 | 0.981 | 0.021 |
| | MagicM | 1054.4 | 0.978 | 0.100 | 1719.4 | 0.985 | 0.074 |
| | Ours | **837.0** | **0.983** | **0.031** | **1144.8** | **0.994** | **0.017** |
| | | FVD↓ | Consistency↑ | TrajSIM↑ | FVD↓ | Consistency↑ | TrajSIM↑ |
| Unaligned | DAS | 773.9 | 0.979 | 0.861 | 2716.3 | 0.992 | 0.656 |
| | GoFlow | 1050.5 | **0.973** | 0.808 | 2978.3 | 0.991 | 0.704 |
| | Ours | **622.3** | 0.976 | **0.908** | **2654.2** | **0.993** | **0.757** |

## 5 CONCLUSION

We introduced FlexTraj, a unified framework for video generation with multi-granularity, alignment-agnostic trajectory control. By encoding segmentation, correspondence, and optional appearance cues into a unified compact representation, FlexTraj overcomes the fragmentation of task-specific conditions. Our efficient sequence-concatenation strategy enables effective conditioning, while the annealing curriculum promotes robust generalization across dense, sparse, and unaligned supervision. Extensive experiments show that FlexTraj not only advances controllability but also broadens the practical applicability of diffusion-based video generation.

## REPRODUCIBILITY STATEMENT

We have made every effort to ensure our work is reproducible. The core technical components (trajectory representation, condition injection, and training strategy) are detailed in §3. Additional details of the implementation can be found in A.1. To further facilitate replication, we report comparative results on DAVIS in Tab. 3 and Tab. 1. Finally, we will publicly release the source code and corresponding model weights upon publication.

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

# A  APPENDIX

## A.1  EXPERIMENT DETAILS

**Implementation details.** We start by constructing trajectory representations. For real-world videos, we first annotate points by SAM (Ravi et al., 2024) for video segmentation and SpatialTracker (Xiao et al., 2024b) for tracking 4,900 uniformly distributed 3D points. We then project these points onto the 2D plane as videos, where the point size is dynamically adjusted to accommodate different levels of granularity: $h = \lfloor 2s \rfloor$ and $w = \lfloor 3s \rfloor$, where $s = \min\left(sqrt(H/x/1.7),\ 4\right)$ and $x$ denotes grid size. For CG synthetic videos, we render the condition video directly in Blender, where each mesh is treated as an instance and each vertex serves as a tracking point.

After constructing trajectory representations, we next describe our annealing training schedule, which consists of four stages: a complete stage of 1,200 steps, a dense stage of 2,400 steps, a sparse stage of 14,000 steps, and finally an unaligned stage of 4,000 steps. We set $p_c$ to 0.5, while $p_s$ and $p_t$ take values in the range $[0, 1]$. The learning rate is fixed at $1 \times 10^{-4}$ for the aligned stages (first three) and reduced to $1 \times 10^{-5}$ for the unaligned stage.

Our model is fine-tuned on the recent video diffusion model CogVideoX-5B I2V (Yang et al., 2024b), which is based on the MM-DiT architecture (Esser et al., 2024). Fine-tuning is performed with LoRA (rank 128, batch size 1) applied to the self-attention query, key, and value projections. Training requires about one week on 8 NVIDIA H800 GPUs, and inference takes roughly five minutes per video when using KV-cache.

**Evaluation dataset.** For evaluation, we use DAVIS (Pont-Tuset et al., 2017) as the standard benchmark and configure it for four tasks: dense, spatially sparse, temporally sparse, and unaligned. Spatial sparsity is simulated by randomly sampling 10 points, while temporal sparsity is obtained by uniformly sampling 2–4 frames from the full sequence. The unaligned setting is generated by randomly jittering the condition videos: we first resize the video to a larger resolution and then crop it back to the target size to obtain tracking points. In addition, we curate FlexBench, which includes 10 videos for each task, half collected from online sources and half synthesized with Blender, to demonstrate applicability for both general and professional users. All videos are trimmed to 49 frames and cropped to $720 \times 480$.

**Metrics.** We employ several standard metrics. For overall video quality, we report Fréchet Video Distance (FVD (Unterthiner et al., 2018)) and Frame Consistency (Esser et al., 2023), which measures CLIP similarity (Radford et al., 2021) between consecutive frames. For motion controllability, we use Trajectory Error (TrajError) (Wu et al., 2024), defined as the average Euclidean distance between trajectories extracted from the generated video and their matched trajectories extracted from the source video. For the unaligned setting, we adopt Trajectory Similarity (TrajSIM) (Pondaven et al., 2025), computed as the mean cosine similarity between the displacement directions of each extracted trajectory in generated video and its closest counterpart in the source video.

**Baseline.** We compare with the most relevant methods for each task. We provide a capability comparison of controllable I2V methods on Tab 2.

Table 2: Comparison of Controllable I2V Methods. (SS: Spatially Sparse, TS: Temporally Sparse)

| Controllable Methods | Point-Traj | 3D-aware | Dense | SS | TS | Unaligned |
|---|---|---|---|---|---|---|
| Diffusion-As-Shader | ✓ | ✓ | ✓ | | | |
| Go-with-the-flow | ✓ | | ✓ | | | |
| MagicMotion | | | ✓ | ✓ | ✓ | |
| Levitor | ✓ | ✓ | | ✓ | | |
| ToRA | ✓ | | | ✓ | | |
| SparseCtrl | | | | | ✓ | |
| Ours | ✓ | ✓ | ✓ | ✓ | ✓ | ✓ |

## A.2  THE USE OF LARGE LANGUAGE MODELS

We used large language models to correct grammar and refine wording for a more formal, academic style during the writing process.

Table 3: Quantitative results for ablation. Our method has the lowest TrajError on the aligned tasks (Dense, Spatially Sparse, Temporally Sparse) and also highest TrajSIM on the unaligned task.

| Ablation | Method | TrajErr↓ (Aligned Tasks) | | TrajSIM↑ (Unaligned Task) | |
|---|---|---|---|---|---|
| | | DAVIS | FlexBench | DAVIS | FlexBench |
| Trajectory Representation | \w CorrID | 0.029 | 0.018 | 0.904 | 0.729 |
| | \w SegID | 0.040 | 0.019 | 0.895 | 0.732 |
| Injection Control | ControlNet | 0.131 | 0.084 | 0.556 | 0.539 |
| Training Strategy | RandomMix | 0.126 | 0.084 | 0.588 | 0.523 |
| | Sparse2dense | 0.126 | 0.094 | 0.592 | 0.662 |
| Final Model | Ours | **0.024** | **0.016** | **0.908** | **0.757** |

## A.3 ANALYSIS ON TRAJECTORY REPRESENTATION METHODS

Existing trajectory representations can be broadly grouped into three categories. *Gaussian map–based methods* (Wu et al., 2024; Yin et al., 2023; Zhang et al., 2025b) enlarge point features with a local radius, which allows capturing neighborhood context. However, they lack explicit temporal correspondence. *Color-propagation approaches* such as DAS (Gu et al., 2025) establish temporal correspondence by propagating colors defined on the first frame, but they cannot represent points that appear later, nor do they encode segmentation information. *Random-embedding vectors methods* (Geng et al., 2025) offer flexibility to model newly appearing points and maintain temporal correspondences, yet they still omit segmentation and appearance cues. In contrast, our method accommodates new points and encodes comprehensive information, including correspondence, segmentation, and optional color.

## A.4 ABLATION STUDY

**Trajectory Representation.** Our representation combines SegID, TrajID, and optional color. Without SegID, there is no distinction on different instances, e.g. two people entering from opposite sides are mistakenly placed together in Fig. 8 (b). Without color, instances are separated correctly but appearance deviates, whereas adding it restores fidelity. On the other hand, motion interpolation becomes ambiguous without TrajID: although the overall shape is preserved, point correspondences are mismatched, causing the pinwheel to rotate incorrectly in Fig. 8 (a).

**Condition Injection Scheme.** A ControlNet-style scheme shows limited control capacity under our training setting: for example, the turtle's head in Fig. 8 (c) fails to rotate to the intended direction, whereas our injection produces accurate motion.

**Training strategy.** We further analyze training strategies. When adopting random mixing or reversed-dense schedule, motion control performance drops. As shown in Figure 8 (d), the girl's head fails to rotate to the correct orientation. In contrast, our annealing strategy maintains stable optimization and ensures precise motion alignment.

**Quantitative Result.** Our method yields the lowest TrajErr on aligned tasks (Dense, Spatially Sparse, Temporally Sparse) and the highest TrajSIM on the unaligned task, confirming both precise motion following and robustness.

## A.5 MORE RESULTS

We provide additional results in Fig. 10, covering all applications introduced in Fig. 1, to more clearly demonstrate the effectiveness of our method.

## A.6 LIMITATION

Our method faces two main limitations. First, it relies on tracking quality: when tracking fails, regions missed by tracking default to free generation, as in Fig. 9 where the woman's glove is misaligned. Second, it inherits constraints from the underlying video generator, including difficulty with large rotations and limited long-term scene memory. For instance, after a 360° camera orbit, scene quality degrades and the original scene cannot be faithfully recovered. Future work includes exploring the integration of explicit memory mechanisms to enhance long-term scene consistency.

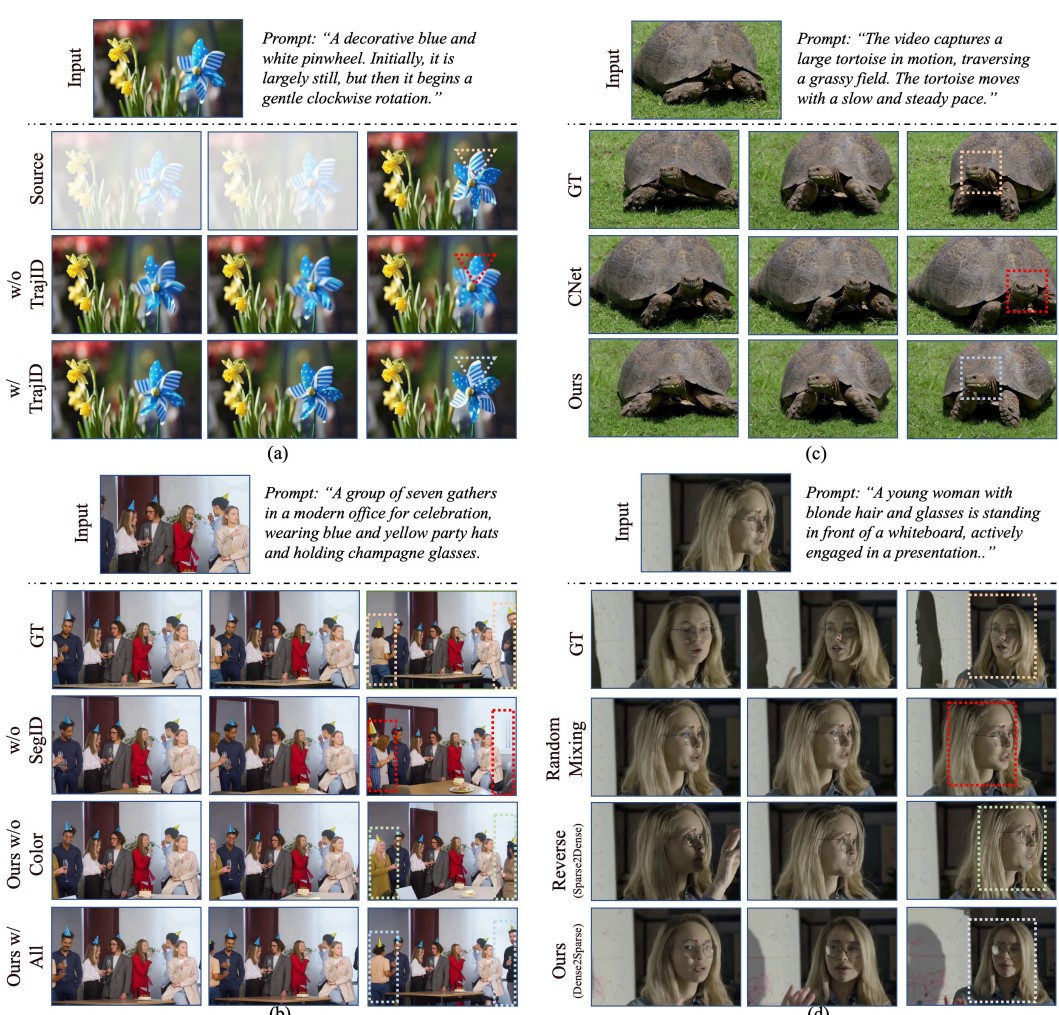

Figure 8: Ablation study examples. (a) *Trajectory representation (TrajID):* without *TrajID*, accurate trajectory control fails due to ambiguous correspondences. (b) *Trajectory representation (SegID and Color):* without *SegID*, newly emerging regions are generated randomly; with *SegID*, they follow the segmentation but lose appearance cues; with all attributes, generation matches the GT. (c) *Condition injection:* ControlNet-style (Zhang et al., 2023) injection provides limited control, whereas ours achieves accurate motion. (d) *Training strategy:* Random mixing or reversed schedules degrade performance, while our annealing strategy preserves accurate alignment.

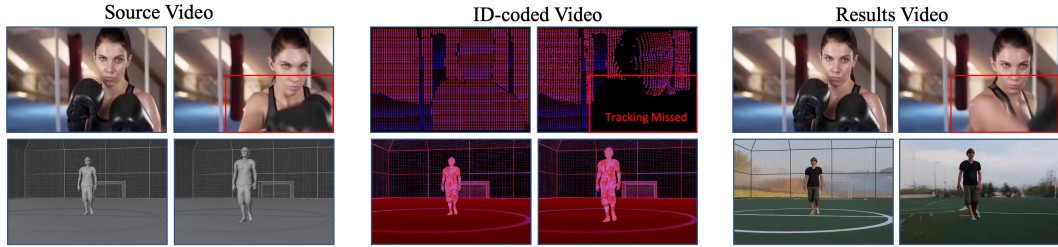

Figure 9: Limitations. Motion alignment is limited by tracking quality (top row: glove), and generation is constrained by the base video model (bottom row: fails on a 360° camera orbit.)

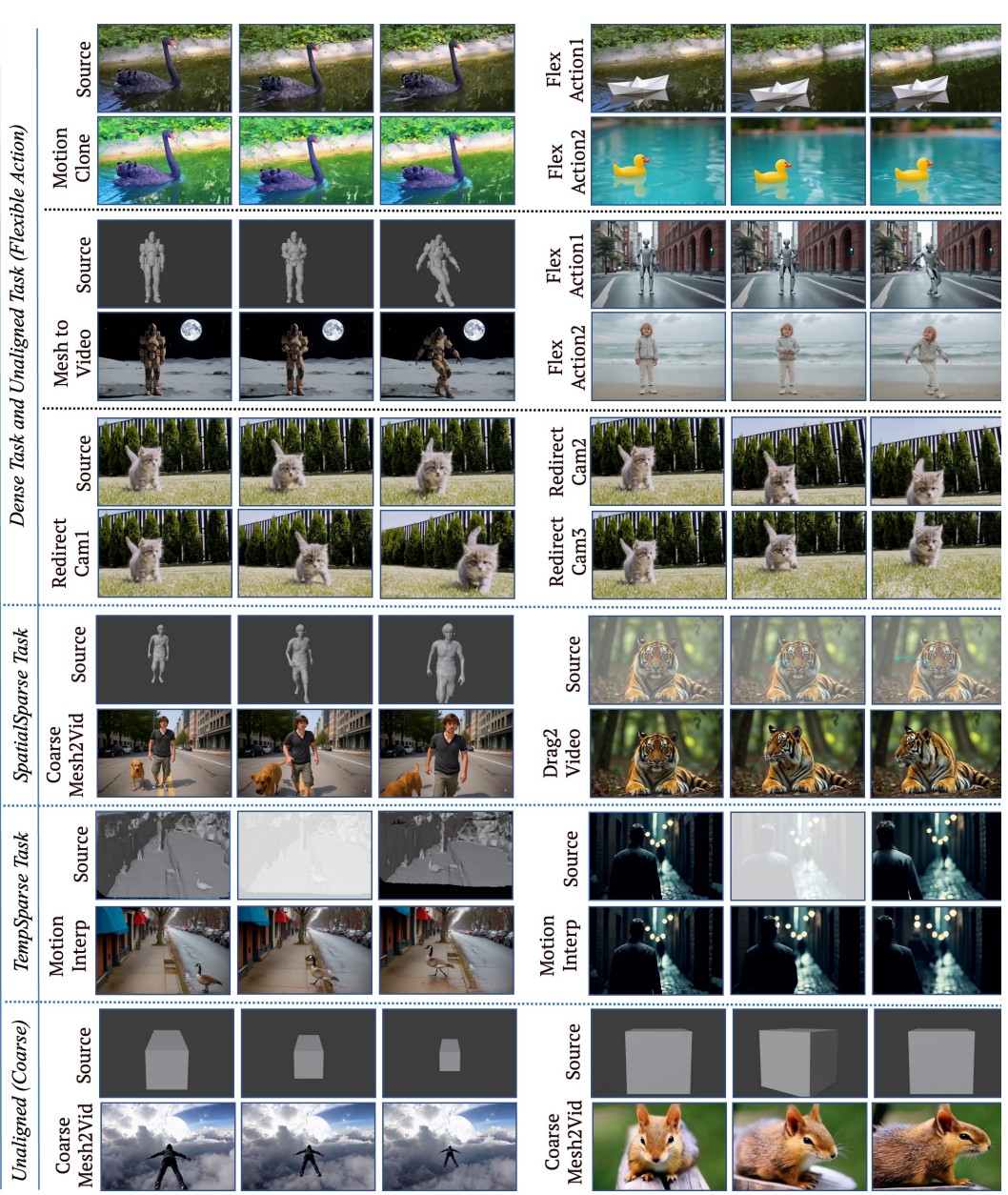

Figure 10: More results. We provide additional results on all the applications here.

