# OpenReview forum: "FlexTraj: Image-to-Video Generation with Flexible Point Trajectory Control"
_ICLR.cc/2026/Conference — ICLR 2026 Conference Withdrawn Submission_

### Official Review · Reviewer_ELNB · 2025-10-31

**Soundness:** 2
**Presentation:** 2
**Contribution:** 2
**Rating:** 2
**Confidence:** 5

**Summary:**

The paper presents an image-to-video generation framework based on point trajectory control. In detail, the proposed FlexTraj leverages a unified point-based motion representation which contains the information of segmentation ID, temporal consistent trajectory ID and color cues. The combined conditions are formed as video and encoded by video VAE. FlexTraj adopts a sequence-concatenation scheme to inject the condition into video diffusion and proposes an annealing training strategy to adapt the model for motion control with different trajectory conditions. Comprehensive experiments conducted on DAVIS verify the efficacy of motion control under the condition of dense, spatially sparse, temporally sparse and unaligned trajectories.

**Strengths:**

1.	The support of multi-granularity and alignment-agnostic trajectory control can be widely applied into various scenarios. The value in the engineering field is high.
2.	Good performances are achieved on the DAVIS and the proposed FlexBench compared to SOTA trajectory control methods.
3.	The generalization ability for unaligned control is impressive which could have much potential for the motion transfer.

**Weaknesses:**

1.	Although the proposed approach supports various motion control scenarios, the technical contribution is limited in my personal opinion. This work is more like an engineering design instead of a research investigation. The point control is widely explored in many previous motion control methods such as DragNUWA and DragAnything. Meanwhile, the sequence concatenation for condition injection in DiT is not new even though it is an effective and efficiency way in real-world implementation. The causal mask and KV cache have no strong relation with the whole network design and the technical is also leveraged from EasyControl.
2.	The technical details about the SegID and TrackID are not clear. From the Figure 2, it is very hard for me to figure out what is the trajectory id and the segmentation id. What is the difference between them? There is no intuitive demonstration for these. Meanwhile, the condition videos shown in Figure 2 is very confusing and there is no meaningful thing. Is the index related with the object instance? Is there any detection or segmentation techniques employed for the target? I have no idea and very confused about these concepts. Nevertheless, I think these concepts are very important sinch they attribute the work a lot for the successful trajectory control.
3.	In the experimental section, most of the baselines are constructed on the UNet. The authors usually claim that the inferior results are caused by the unpowerful architecture. As such, I think the comparison are not fair and the contribution can be brought by the stronger video DiT models.
4.	There should be more quantitative analysis for ablation studies rather than only show the visual cases. It is not convincing in my opinion.

**Questions:**

Please see the weakness.

---

### Official Review · Reviewer_fZAa · 2025-10-31

**Soundness:** 3
**Presentation:** 3
**Contribution:** 2
**Rating:** 4
**Confidence:** 4

**Summary:**

The paper introduces FlexTraj for image-to-video (I2V) generation that allows for precise and flexible control over object motion using user-defined point trajectories. FlexTraj couples point trajectories with contextual information like segmentation masks and appearance attributes. This multi-granularity representation is integrated into a pre-trained I2V diffusion model via a tailored condition injection mechanism and an optimized annealing training strategy. The results show an advancement in controllable video generation.

**Strengths:**

- The combined use of segmentation IDs, temporal IDs, and optional appearance cues is effective.
- The comparison against ControlNet-style injection indicates that the proposed condition injection method is effective.
- The annealing training strategy is shown to be crucial for preserving the base model's coherence while ensuring alignment to the control signal.
- The paper is overall well-written and easy for readers to follow.

**Weaknesses:**

- The paper claimed that FlexTraj is the first framework to support multi-granularity and alignment-agnostic trajectory control. However, in fact, the video generation with 3D point trajectory control has been well studied in previous works, e.g., Diffusion-As-Shader (DAS). And the control signals mentioned in the paper are all variations of 3D point trajectories, which can be obtained through a simple conversion. Therefore, I don't think this can be considered a major contribution.
- The data processing pipeline used by FlexTraj is highly similar to that of DAS. Furthermore, I believe the methodology involves a combination of several existing and commonly used techniques. i.e., token concatenation and LoRA structures have been widely used and proven effective in controllable generation tasks, causal mask and KV cache are also borrowed from EasyControl. Claiming the efficient sequence-concatenation strategy as a major contribution is relatively weak.
- The generative capacity remains constrained by the pre-trained base I2V model. Tasks requiring significant outpainting, strict physical constraints, or substantial changes to the scene structure may still be challenging.
- Some highly related published papers are missing from the related work section.

Trajectory Attention For Fine-grained Video Motion Control, ICLR 2025

GS-DiT: Advancing Video Generation with Dynamic 3D Gaussian Fields through Efficient Dense 3D Point Tracking, CVPR 2025

GEN3C: 3D-Informed World-Consistent Video Generation with Precise Camera Control, CVPR 2025

**Questions:**

- What about the performance of FlexTraj when applied to highly non-rigid phenomena, such as fluid dynamics, e.g., water, smoke, fire, or deformable materials, e.g., cloth, while point tracking and segmentation may fail?
- How does the proposed trajectory representation specifically handle points or segments that become occluded and then re-emerge later in the video? Similarly, how does FlexTraj handle newly appearing objects?

---

### Official Review · Reviewer_2vtv · 2025-11-01

**Soundness:** 3
**Presentation:** 3
**Contribution:** 3
**Rating:** 6
**Confidence:** 4

**Summary:**

This paper presents *FlexTraj*, a unified framework for image-to-video generation that introduces flexible point trajectory control. The method encodes motion as a set of annotated 3D points—each carrying segmentation, trajectory, and optional color attributes—enabling both dense and sparse motion control. A core contribution lies in its *efficient sequence-concatenation* scheme for condition injection, which improves controllability and supports unaligned conditions. The authors further propose a density and alignment *annealing training* strategy, allowing the model to generalize from dense aligned conditions to sparse and unaligned scenarios. Experiments demonstrate consistent improvements in motion controllability and visual quality over a wide range of baselines.

**Strengths:**

1. Provides a **comprehensive and unified** approach to motion control across multiple levels of granularity (dense, sparse, unaligned).

2. The **point-based representation** is elegant and general, bridging various types of control signals in a compact and interpretable form.

3. The **efficient sequence-concatenation mechanism** is well-motivated and empirically validated, offering practical benefits in training stability and inference speed.

4. The **annealing curriculum** for handling condition sparsity and misalignment is a thoughtful design and contributes to strong robustness.

**Weaknesses:**

1. The framework seems **computationally heavy**, and real-time feasibility or scalability to longer sequences is not discussed.

2.  Occasional lack of **theoretical depth**—most claims are empirically observed without deeper analysis (e.g., why sequence concatenation generalizes better).

3. Computational overhead is mentioned briefly, but training efficiency and inference scalability deserve more quantitative reporting to assess practical feasibility.

**Questions:**

See Weakness

---

### Official Review · Reviewer_nZYY · 2025-11-01

**Soundness:** 3
**Presentation:** 3
**Contribution:** 2
**Rating:** 4
**Confidence:** 4

**Summary:**

This paper presents FlexTraj, a point-trajectory-based controllable image-to-video framework. The method introduces a unified representation that encodes segmentation IDs, temporal IDs, and optional color cues, and feeds these as tokens into a video diffusion model (CogVideoX-5B) via sequence concatenation with LoRA and a causal mask, similar to EasyControl. The authors further propose a data annealing curriculum to handle dense, sparse, and unaligned inputs. Experiments show improvements over point-trajectory and structure-guided baselines across dense, sparse, temporal, and unaligned settings, with strong motion controllability and competitive visual quality

**Strengths:**

1.The paper introduces a unified point-trajectory conditioning interface that supports dense, sparse, and unaligned trajectory inputs, offering good flexibility and applicability across diverse video control scenarios.

2.The proposed training strategy that gradually anneals data density and alignment quality helps the model stably learn from heterogeneous trajectory formats, enabling consistent motion control under varied supervision signals.

3.The method is evaluated on DAVIS, FlexBench, and other benchmarks, and the results demonstrate strong motion controllability and competitive visual quality. The system is well-engineered and shows a high level of completeness.

**Weaknesses:**

1. Limited technical novelty: The core design largely extends EasyControl’s token-concatenation and conditioning mechanism to video without introducing new temporal control modeling or control representations. The contribution is mostly an application-level adaptation rather than a fundamental methodological advance for controllable video generation.


2.Lack of direct comparison with classical controllable video baselines such as VACE under identical data and training settings. Since EasyControl and VACE are both important backbones in this area, a fair head-to-head evaluation is necessary to convincingly support the claimed improvements in controllability and quality.


3.EasyControl’s causal attention mechanism is designed specifically to support multiple control types simultaneously. However, this paper only demonstrates single-control scenarios, failing to reveal the benefit of causal attention or justify its necessity in this setting. As a result, it remains unclear whether the method extends beyond a straightforward single-control EasyControl adaptation.

**Questions:**

What is the motivation for using causal attention? In controllable video generation tasks, what advantages does it offer compared to a bidirectional attention design?

---

### Note · Authors · 2025-11-12

I have read and agree with the venue's withdrawal policy on behalf of myself and my co-authors.